# Seat-to-Head Transmissibility Responses of Seated Human Body Coupled with Visco-Elastic Seats

**K. N. Dewangan [1], Yumeng Yao [2],\* and S. Rakheja [3]**

[1] Department of Agricultural Engineering, NERIST, Nirjuli 791109, India
[2] School of Mechanical Engineering, University of Shanghai for Science and Technology, Shanghai 200093, China
[3] CONCAVE Research Center, Concordia University, Montreal, QC H3G 1M8, Canada
\* Correspondence: yu_yao@usst.edu.cn

**Abstract:** This study investigated the seat-to-head vibration transmissibility (STHT) responses of 58 subjects (31 males and 27 females) seated on three different elastic seats with (WB) and without back support (NB) and under three levels of vertical vibration (0.25, 0.50 and 0.75 m/s$^2$ RMS) in the 0.50–20 Hz range. The STHT responses with elastic seats were significantly different from the widely reported responses with rigid seats, irrespective of sitting and excitation conditions. The peak STHT magnitudes with elastic seats were relatively higher than those obtained with a rigid seat. Moreover, the transmission of seat vibration showed a strong dependence on the elastic properties of the body-seat coupling. The primary resonance frequencies were also significantly different among the elastic seats. Compared to NB conditions, the peak STHT magnitudes and the primary resonance frequencies obtained with WB conditions were significantly lower. An increase in excitation magnitude resulted in a statistically significant ($p < 0.001$) decrease in the primary resonance frequency.

**Keywords:** seated body head vibration; through-the-body biodynamic response; elastic seats; pelvis orientation and rotation; sitting condition



## 1. Introduction

The flow of whole-body vibration (WBV) to a seated human occupant is generally characterized in terms of the transmissibility of seat vibration to different body segments, such as the vertebra, pelvis, thorax, shoulder and head. Such through-the-body frequency response functions permit a better understanding of the participating modes of vibration, which further provide knowledge on potential adverse health effects of WBV exposure [1–4]. The measurements of the through-the-body biodynamic response functions, however, have been deemed challenging in terms of notable errors due to skin movement over bones and the lack of reliable in vitro measurement methods. These are also evident from substantial inter-subject variances in measured vibration responses [5–7]. Among the different segments' vibration responses, the STHT response has been most widely investigated because of the relative ease of its measurement and lower errors due to skin movement. Studies have shown substantially lower inter-subject variances in measured head responses compared to those of the spinal segments [8]. Transmission of vertical seat vibration to the occupants' head has been extensively investigated in the sagittal plane under widely different posture-, excitation- and seat geometry-related experimental conditions [9–11].

A review of studies reporting STHT response to WBV [12] and a synthesis of selected datasets [10] suggest considerable differences among the reported results. These have been associated with board differences in experimental conditions considered in individual studies. Moreover, individual studies have shown large inter-subject variability in measured STHT responses, which are partly due to differences in the physical characteristics of participants [7,13]. Despite the observed variabilities, the reported STHT responses to vertical WBV have provided considerable knowledge on the mechanical properties of the body,

which facilitated formulations of analytical models for seating design applications [6,14,15]. The measured responses consistently exhibit dominant peaks in the 4–6 Hz frequency range, which is considered the fundamental vibration mode of the seated human body. It has been suggested that the primary resonance frequency may be related to bending in the lumbar spine caused by the rocking of the pelvis [16], while Zheng et al., [17] opined that it is relevant to head motion, the spinal column and the pelvis, as well as a bending mode of the upper thoracic and the cervical spine.

Characterizations of the transmission of seat vibration to the seated occupant's head have been mostly limited to the body seated on a rigid platform, with only a few exceptions [7]. This condition not only facilitated the measurement of vibration at the body-platform interface, but also permitted the study of body response behavior uncoupled from the seat. Polyurethane foams (PUF), widely used in automotive seats, exhibit non-linear visco-elastic behavior [14]. Therefore, the contributions of body coupling with visco-elastic seats to the STHT responses have not been adequately explored. Visco-elastic properties and seat contours play an important role in the body-seat contact area and the distribution of contact force, which are further influenced by many physical factors in addition to sitting posture [18,19]. Compared to elastic seats, a rigid seat yields a considerably smaller body-seat contact area, and thereby, a substantially higher localized peak contact pressure. Therefore, body-coupling with an elastic seat can yield important effects on the vibration biodynamics of the seated body. Wu et al. [20,21] compared contact area and mean contact pressure characteristics of elastic and rigid seats. The study revealed a substantially higher contact area and more even contact pressure on the elastic seat than that on the rigid seat. This tendency was also observed by Dewangan et al. [22,23].

A seated body supported on an elastic seat also yields a notably higher pelvic orientation when compared with a rigid seat. The variations in the pelvic orientation may affect vibration transmission to the head. Koo et al. [24] reported notably different pelvic tilts for the PUF and Roho cushions, considering six different sitting postures. Pelvis orientation and rotation also depend on the stiffness of the seat cushion [25]. Relatively soft air cushions may cause relatively higher pelvic tilt due to uneven deformations in the presence of localized contact pressure peaks. Moreover, Lemerle and Boulanger [26] observed notable variations in the pelvic rotation of humans coupled with a suspension seat under vertical vibration excitation. The variations in the pelvic orientation, when seated on an elastic seat, may also affect vibration transmission to the head, although the effects have not yet been explored.

The static stiffness of PUF seat cushions tends to vary with seated body load, often denoted as the seat preload. The dynamic stiffness of the PUF cushion, on the other hand, varies with the magnitude and frequency of vibration [14,27]. Therefore, the body mass, magnitude and frequency of seat vibration yield couple effects on the nature of seat vibration transmission to the occupant's head. Pope et al. [28] measured vibration transmissibility at L3 and at the head with a bite bar for three female subjects seated on three different types of cushions, while exposed to impacts in the vertical direction. The softer cushion revealed higher magnitudes of peak transmitted vibration, but a lower fundamental natural frequency compared to the stiffer cushion. Hinz et al. [13,29] measured the vertical STHT responses of humans sitting on a hard seat and on a suspended seat. The vibration transmissibility, however, was defined with respect to the vibration at the seat base as opposed to the vibration at the body-seat interface, which would be different for the two seating conditions. Both studies considered 39 subjects seated without back support. The magnitude of frequency-weighted transmitted vibration was higher for the suspended seat (0.70 m/s$^2$) compared with the rigid seat (0.60 m/s$^2$). This may in part be due to differences in vibration at the human-seat interface of the two seats, although STHT is known to be less sensitive to the magnitude of vibration excitation [7]. Hinz et al. [29] suggested that seat surface quality, sitting posture and the use of the backrest should be considered important factors while studying vibration transmission to the head of an occupant sitting on an elastic seat.

The aforementioned studies have revealed that the elasticity of the seat can significantly alter the body-seat contact properties and pelvic orientation, while only limited knowledge exists on its effect on STHT responses. In the present study, an experimental methodology is developed to measure the STHT responses to vertical seat vibration when the body is coupled with three different visco-elastic seats. The measurements are conducted with 58 participants (31 males and 27 females) seated on three different elastic seats considering two sitting conditions (NB–without back support; WB –vertical back support) and three excitation magnitudes (overall RMS acceleration: 0.25, 0.5, 0.75 m/s$^2$) in the 0.5–20 Hz frequency range. The STHT responses are compared with those reported for a rigid seat to highlight the effects of elastic seats. The influences of the elastic properties of seats and sitting conditions on the STHT responses are further discussed.

## 2. Measurement and Analysis Methods

### 2.1. Participants and Seat Characteristics

The study employed 31 males and 27 females who had never experienced any musculoskeletal or cardiovascular symptoms. The experimental setup, procedure and safety guidelines were explained to each subject. Each participant consented to the experimental protocol, which had been approved by the Human Research Ethics Committee of Concordia University, and ethical guidelines were followed in the experiments. Anthropometric data in terms of stature, body mass and sitting height were measured for each subject prior to the experiments. The relevant information of the participants is shown in Table 1.

**Table 1.** Range (minima, maxima, means and standard deviations) of anthropometric dimensions of the participants.

| Particulars | Minimum, Maximum, Mean (Standard Deviation) | | |
| | Male (*n* = 31) | Female (*n* = 27) | All (*n* = 58) |
| --- | --- | --- | --- |
| Age (years) | 23.0, 58.0, 31.2 (7.2) | 19.0, 49.0, 19.0 (7.1) | 19.0, 58.0 25.5 (7.1) |
| Stature (m) | 1.59, 1.92, 1.75 (0.08) | 1.48, 1.73, 1.63 (0.07) | 1.48, 1.92, 1.69 (0.09) |
| Sitting height (cm) | 81.3, 96.7, 88.8 (6.2) | 63.2, 88.3, 81.0 (7.7) | 63.2, 96.7, 85.2 (6.3) |
| Body mass (kg) | 55.0, 106.0, 79.8 (15.7) | 45.5, 72.5, 60.1 (8.3) | 45.5, 106.0, 70.6 (15.4) |
| Body mass index (kg/m$^2$) | 19.96, 34.99, 26.12 (4.24) | 15.78, 26.31, 22.52 (2.73) | 15.78, 34.99, 24.44 (3.87) |
| Seat pan contact area (cm$^2$) | 211, 1050, 575 (195) | 250, 890, 515 (175) | 211, 1050, 547 (185) |
| Mean contact pressure (kPa) | 8.1, 26.7, 13.5 (4.6) | 5.9, 14.0, 9.5 (2.3) | 5.9, 26.7, 11.6 (3.9) |

*n*: number of participants.

Three elastic seats were selected for the study, namely, a flat 8 cm thick PUF block cushion with leather covering (seat A), a contoured automotive PUF seat cushion (seat B) and an inflatable air-bubble cushion (seat C), as shown in Figure 1a–c. The force-deflection characteristics of all the elastic seats were measured in the laboratory using a 200 mm diameter loading indenter [30]. The procedure for force-deflection measurements is provided in detail by Dewangan et al. [23]. It should be noted that the standardized indenter load is predominantly applied to the ischium region, which supports about 55% of the standing body masses [20]. The measured fore-displacement data were used to derive the static stiffness of each seat for three different ischium loads, 330, 440 and 530 N, which correspond to standing body masses of 60, 80 and 96 kg, respectively.

### 2.2. Experimental Setup and Measurement Methods

A Whole-Body Vertical Vibration Simulator (WBVVS), as described in M-Pranesh et al. [8], was used to synthesize the desired seat vibration and measure seat-to-head vibration transmission (STHT) characteristics of the participants. A rigid seat with a horizontal seat pan (449 × 456 mm) and a vertical backrest were mounted on the platform supported by two servo-hydraulic actuators. The height of the seat pan was 445 mm. In order to simulate a driving posture, a steering wheel and a steering column were fixed on the platform. The vertical acceleration at the rigid seat was measured via a single-axis accelerometer

(B&K 4370) placed on the bottom face of the seat pan. Measurements with elastic seats were performed by positioning each selected cushion on the rigid seat pan. The height of the seat pan, however, was adjusted to ensure an identical sitting height for each elastic seat. Two tri-axes micro-accelerometers (ADXL 330, 14 mm × 14 mm; 1.4 mm thick) were placed on the cushion surface using Douglass tape so that ischial tuberosities of the seated participant were near these accelerometers. Figure 1d illustrates the placement of the accelerometers on seat A. The initial measurements were also performed with a standardized seat-pad accelerometer [31] to examine the validity of the micro-accelerometers. The comparison suggested reasonably good agreement between the two methods, as reported by Dewangan et al. [22]. In order to measure the head vibration, a three-axis accelerometer was positioned on an adjustable helmet-strap comprised of a ratchet mechanism [11].

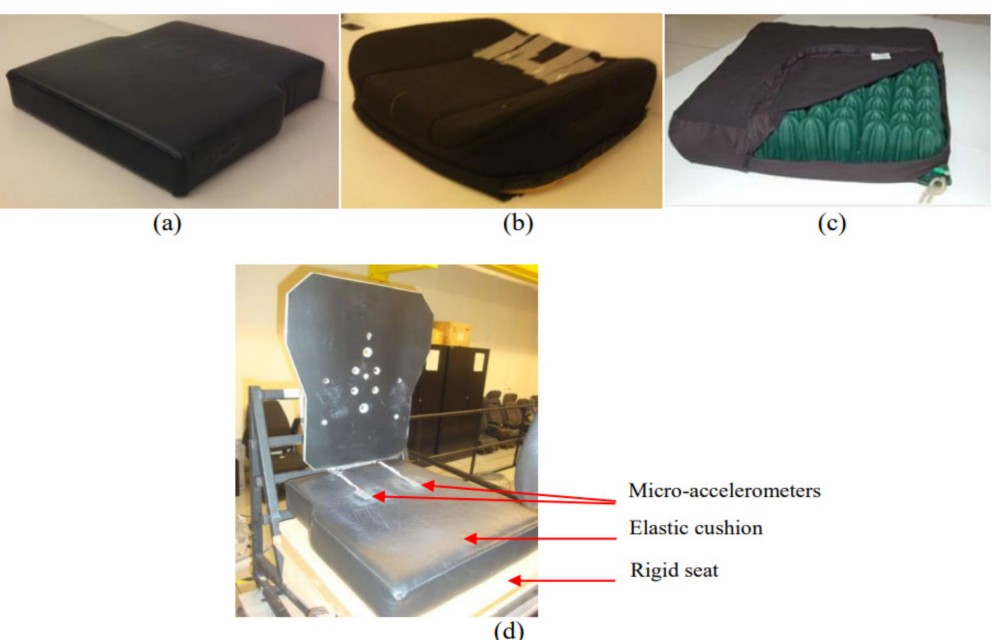

**Figure 1.** Selected cushions and test seat: (**a**) flat PUF (seat A); (**b**) contoured automotive PUF (seat B); (**c**) inflatable air-bubble cushion (seat C); and (**d**) placement of accelerometers on seat cushion A.

Three different levels (0.25, 0.50 and 0.75 m/s² RMS) of white noise random vibration in the frequency range of 0.5–20 Hz with nearly flat acceleration PSD were synthesized for the study. It should be noted that the elastic seat can significantly alter the transmission of vibration excitation at the seat base [32]. The characterization of seat-to-head vibration transmissibility (STHT) responses, however, necessitates identical levels of seat vibration for all the elastic seats in order to study the effects of elastic seats, irrespective of the seated body mass. Therefore, the average of the two micro-accelerometer signals was treated as the feedback to the vibration controller (VR 9500) for synthesizing the desired vibration spectra at the body-seat interface. Considering that the vibration transmissibility characteristics of an elastic seat greatly vary with the characteristics of the seat, as well as the body mass supported on the seat, the participants were grouped into three subsets based on standing body mass (mean mass ≈ 55, 81 and 96 kg). A total of 27 drive files were subsequently synthesized to realize three desired vibration accelerations for the three elastic seats and three participant groups.

A seat pressure sensing mat fabricated by Tekscan (range: 207 kPa; resolution: 0.83 kPa) was placed on the elastic seat to measure the contact area and pressure at the interface of the participant and the seat. Each participant was required to sit on the elastic seat without back support and put his/her hands on the steering wheel. An adjustable feet support was used to support the nearly vertical lower legs and horizontal thighs of the subject. The seat contact area and contact pressure data were recorded under static sitting conditions

for 10 s. The participant was then asked to wear the helmet-strap head accelerometer and adjust its tension to ensure a tight but comfortable fit. The experimenter made the necessary adjustments to ensure the appropriate orientation of the head accelerometer to align with the fore-aft and vertical axes. The participant was asked to sit upright with his/her hands on the steering wheel and look at an object at eye level fixed on the facing wall. In order to reduce intra-subject variance, the experimenter visually monitored the posture and head orientation of the participant during the tests. The vertical acceleration of the seat pan and elastic seat, together with the vertical and fore-aft accelerations of the head, were acquired by a multi-channel spectral analysis system using a bandwidth of 50 Hz and a frequency resolution of 0.0625 Hz. Each measurement took 60 s (12 averages considering 75% overlap). Afterward, the measurements were conducted under different vibration levels, back support conditions and elastic seats. For the WB conditions, each subject was advised to maintain a similar posture with the mid-back in contact with the backrest of the seat. Each measurement was repeated twice. The order of participants performing the experiment were randomized so as to avoid learning effects. The participants were requested to relax for 1–2 min between successive trials and for about 15 min between the elastic seats.

### 2.3. Data Analysis

$H_1$ function was used to obtain the vertical and fore-aft seat-to-head vibration responses. Similarly, the acceleration transmissibility of each elastic seat was derived from the seat pan and the mean of two elastic seat acceleration signals. The mean responses of the two trials were evaluated for both vertical and fore-aft STHT responses of each participant for the back support and excitation conditions considered. Subsequently, the mean responses of the participants were attained for the two sitting and three excitation conditions to investigate the effects of elastic seat characteristics on the vertical and fore-aft STHT responses. The minimum, maximum, mean and standard deviation of the measured contact pressure and contact area were also obtained for different back supports and seats, as shown in Figure 2.

Statistical analyses of the measured data were performed with SPSS software version 22.0 and Microsoft Excel 2013. Initially, the Shapiro-Wilk test was performed to check for the normality of measured data. One-way repeated measures analyses of variance (*r*ANOVA) were conducted to evaluate variations in the vertical and fore-aft STHT magnitudes between the selected seats at selected frequencies for different test conditions. Three-way repeated measures ANOVA was further used to assess the statistical significance of main factors (seat, back support condition and excitation magnitude) on the peak magnitudes of the vertical and fore-aft STHT responses and corresponding frequencies. Additionally, correlations between peak vertical and fore-aft magnitude and the corresponding frequency were also investigated.

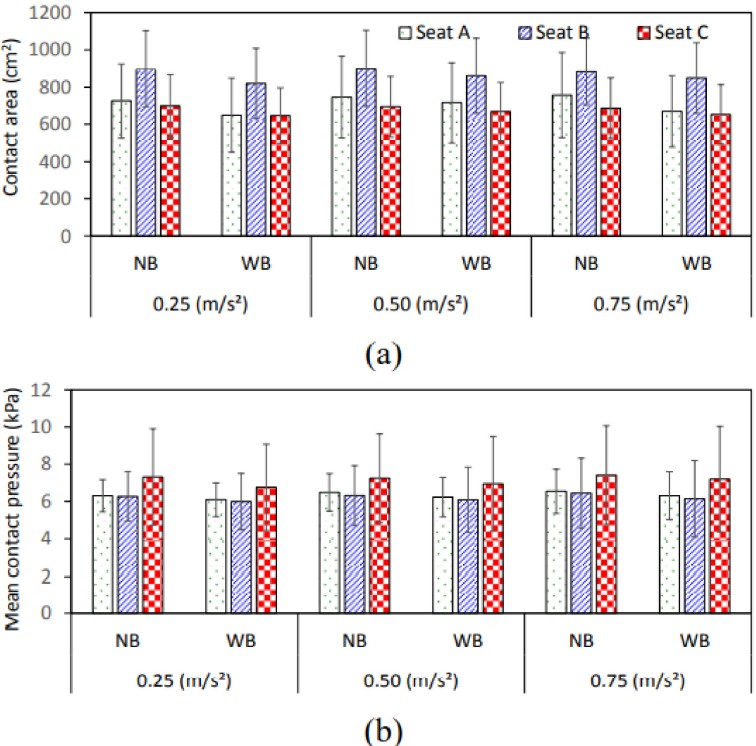

**Figure 2.** Means and standard deviations (shown as error bars) of: (**a**) contact area; and (**b**) contact pressure measured with 58 participants sitting on different elastic seats with two sitting and three excitation conditions.

## 3. Results

### 3.1. Properties of Elastic Seat

Body seat interface contact area and mean contact pressure of the elastic seats were measured for the two sitting and three excitation conditions. Figure 2 shows the mean contact area and mean contact pressure for the three elastic seats together with standard deviations of the means (shown as error bars). The results suggest notable differences in the mean contact area and mean contact pressure obtained for the three seats, separately from large variations. One-way *r*ANOVA showed significant ($p < 0.01$) variation in the contact area and mean contact pressure among the elastic seats. The contact area is considerably high for seat B compared with seat A and seat C. Owing to its lower contact area, the measured mean contact pressure for seat C is higher than those of the PUF seats (seats A and B). The static stiffness of the elastic seats was measured under laboratory conditions, which showed nonlinear characteristics. The stiffness of the elastic seats calculated near three preloads (ischium loads) are presented in Table 2. The results suggest a hardening of the elastic seats with an increase in preload. Seat A exhibits the least static stiffness among all the elastic seats. Seat B is relatively soft under the lower preload (330 N), but is very stiff under higher loads of 440 and 530 N.

**Table 2.** Static stiffness (kN/m) of elastic seats under selected preloads.

| Seats | Preload (N) | | |
|---|---|---|---|
| | 330 | 440 | 530 |
| Seat A | 20.55 | 33.98 | 40.70 |
| Seat B | 21.26 | 40.29 | 52.34 |
| Seat C | 26.20 | 36.09 | 50.51 |

Seat A: flat PUF, Seat B: contoured PUF, Seat C: air cushion.

Vibration transmissibility characteristics of the elastic seats were also evaluated for all 58 participants with two sitting conditions and three excitation magnitudes. As an example, Figure 3 illustrates vibration transmissibility characteristics of elastic seats with participants seated with NB and exposed to 0.50 m/s² RMS excitation. The results show large inter-subject variations in vibration transmissibility characteristics of elastic seats. The resonance frequency of seats A, B and C varied in the range of 4.06–6.38, 4.06–5.63 and 4.06–6.19 Hz, respectively. The coefficient of variation (CoV) of the peak vibration transmissibility magnitude for seats A, B and C occurred in relatively broad ranges, namely, 18–31%, 12–28% and 15–33%, respectively. The vibration transmissibility responses of the seats appear to be strongly dependent on the body mass of the seated participants, in addition to the stiffness property. From the results, it is evident that the elastic seat greatly alters the seat base vibration transmitted to the body-seat interface. The mean vibration transmissibility of the three selected elastic seats are compared in Figure 3d, which suggests higher peak transmissibility for seat C (air cushion) than the PUF seats (seats A and B). Higher peak transmissibility for seat C also suggests lower damping compared to that of the PUF material.

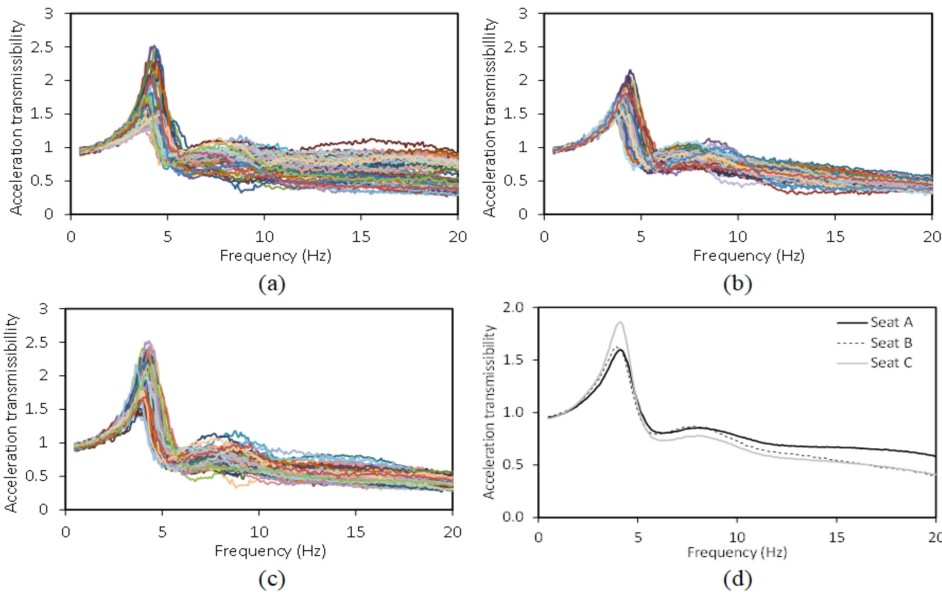

**Figure 3.** Acceleration transmissibility of three seats with 58 participants seated without back support under 0.50 m/s² RMS excitation at the body-seat interface: (**a**) seat A; (**b**) seat B; (**c**) seat C; and (**d**) mean acceleration transmissibility.

*3.2. STHT Response Characteristics*

Figure 4a–c, as an example, illustrates vertical STHT magnitude responses of 58 participants seated on seats A, B and C, respectively, under NB conditions and exposed to 0.50 m/s² RMS acceleration excitation. The results suggest large inter-subject variations in the STHT response, while the primary resonance frequency (frequency corresponding to peak STHT magnitude) occurs in a relatively small range for all the seats. The primary resonance frequency for seats A, B and C for the NB sitting condition varied in the 4.00–6.38, 4.06–5.63 and 4.06–6.19 Hz ranges, respectively. The primary frequencies for the WB sitting condition were observed in the ranges of 3.31–6.06, 4.06–6.19 and 3.81–6.06 Hz, respectively, for the three seats. The CoV of the STHT magnitude for seats A, B and C for NB sitting condition varied from 14–35%, 11–30% and 15–34%, respectively, in the vicinity of the primary resonance frequency. The CoV for the WB sitting condition was slightly lower than the NB sitting condition, which occurred in the 10–30%, 10–26% and 12–28% ranges for seats A, B and C, respectively. A similar degree of scatter and trends were also observed in responses obtained under other test conditions involving different excitation magnitudes. The vertical STHT responses also showed a secondary peak for some subjects in the 8–14 Hz frequency range.

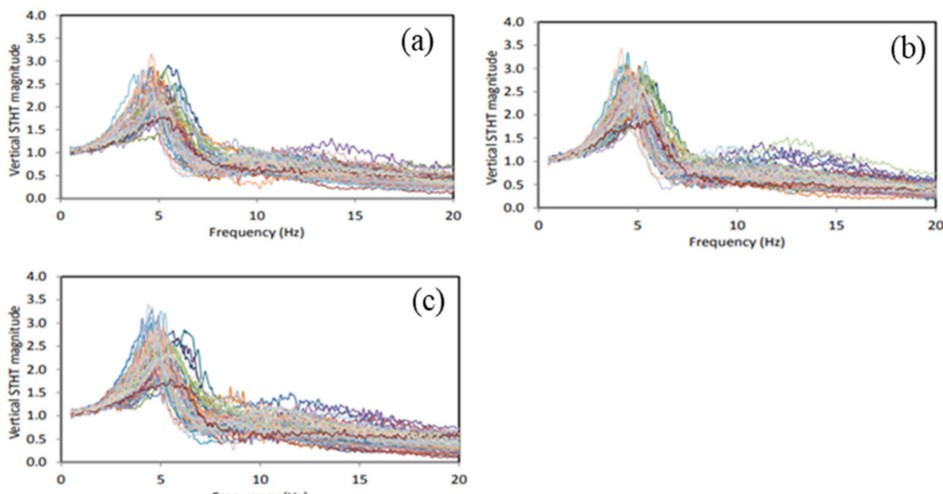

**Figure 4.** Vertical STHT magnitude responses of 58 participants seated without back support under 0.50 m/s$^2$ RMS excitation: (**a**) seat A; (**b**) seat B; and (**c**) seat C.

Figure 5 illustrates the fore-aft STHT responses of 58 participants seated on the elastic seats under the same experimental conditions (NB and exposed to 0.50 m/s$^2$ RMS acceleration excitation). The results show the peak fore-aft STHT magnitudes for seats A, B and C occur in the 3.50–6.00, 3.31–5.81 and 3.31–5.94 Hz frequency ranges, respectively. The corresponding ranges for the WB sitting condition were obtained as 2.94–5.75, 2.81–6.06 and 2.38–6.31 Hz. The fore-aft STHT responses revealed a relatively higher scatter compared with the vertical STHT responses around the primary resonance frequency. For the NB sitting condition, CoVs of peak magnitudes were in the range of 35–46%, 32–39% and 30–44% for seat A, B and C, respectively, while the corresponding ranges for the WB sitting condition were 22–34%, 25–37% and 26–41% at 0.50 m/s$^2$ RMS excitation. Similar to the vertical STHT responses, the fore-aft magnitude data were relatively less scattered with the WB sitting condition compared to the NB sitting condition. In general, the fore-aft magnitude responses for participants sitting without back support also revealed a peak in the 1.5–2 Hz range. A similar degree of scatter and trends were also observed in responses obtained under other laboratory conditions, including different sitting conditions and excitation magnitudes.

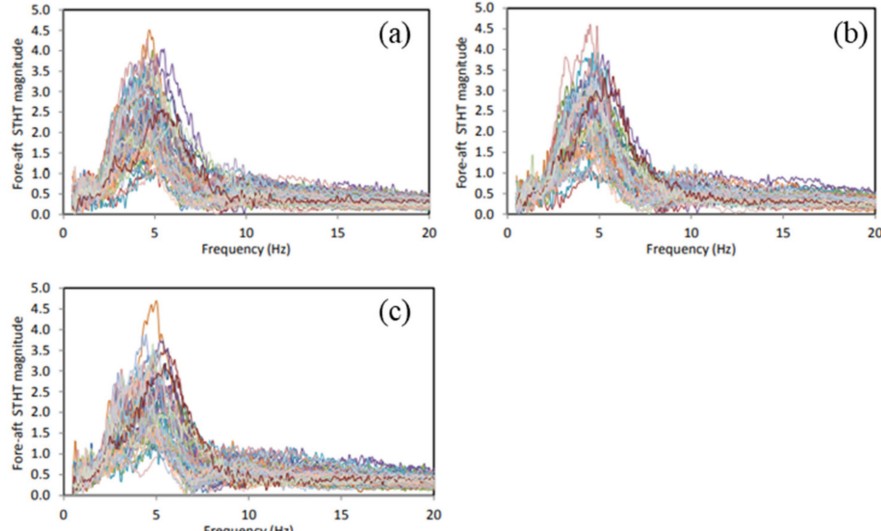

**Figure 5.** Fore-aft STHT magnitude responses of 58 participants seated without back support under 0.50 m/s$^2$ RMS excitation: (**a**) seat A; (**b**) seat B; and (**c**) seat C.

*3.3. Mean Responses*

Figure 6 compares the mean vertical STHT responses of all participants seated on the elastic seats with two sitting and three excitation conditions. The figure also compares the mean responses obtained for the rigid seat under similar sitting and excitation conditions, as reported in [22]. There are important differences in STHT responses obtained for the elastic and rigid seats. Compared to elastic seats, the peak response magnitudes obtained with the rigid seat are notably lower. While the rigid seat yields lower STHT magnitudes of up to 6 Hz for both sitting conditions, the magnitudes tend to be notably higher at frequencies above 10 Hz for the NB condition. The STHT responses obtained for the elastic seats, however, show comparable magnitudes at frequencies below 3 Hz and above 10 Hz, regardless of the sitting and excitation conditions considered. Seat C, however, constitutes an exception under the 0.25 m/s$^2$ RMS excitation.

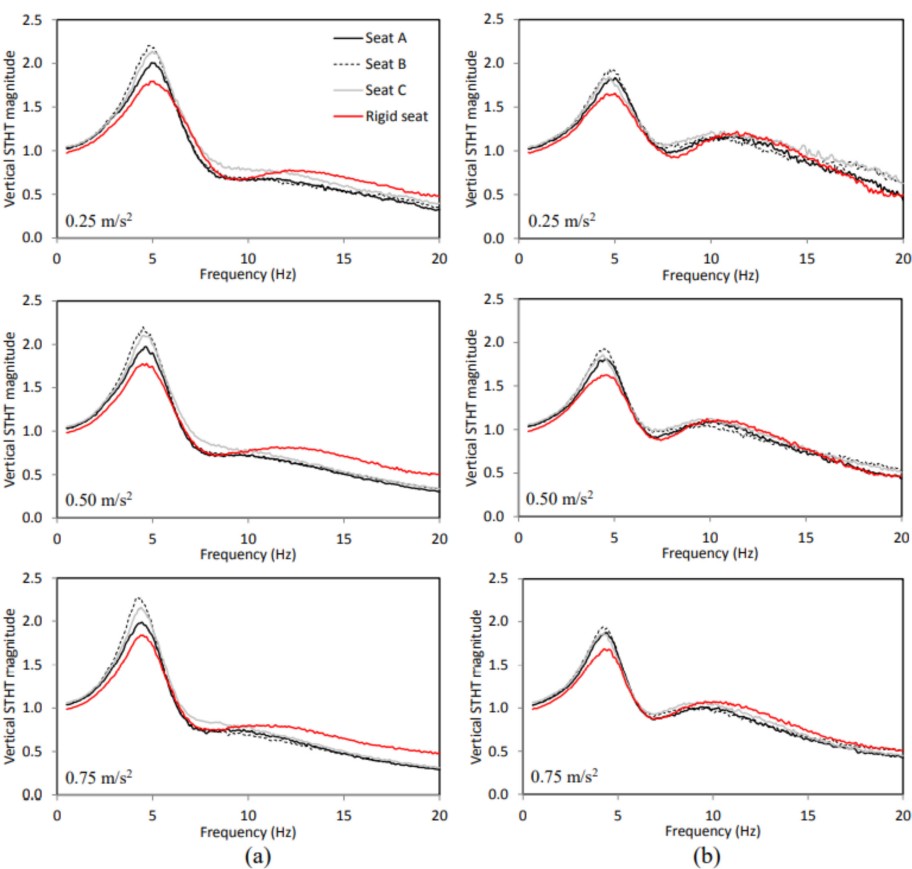

**Figure 6.** Mean vertical STHT magnitude of 58 participants sitting on the elastic and rigid seats under three excitations: (**a**) without back support; and (**b**) with back support.

The STHT peak magnitudes obtained for seat B are slightly higher than those observed for seats A and C, irrespective of the sitting and excitation conditions. The mean peak STHT magnitude obtained with seat A is the lowest among all the seats for the NB sitting condition, as seen in Figure 6. The coupling effect of the sitting condition on the peak STHT magnitude is evident between the seats. The STHT magnitudes obtained with the elastic seats exhibit notable differences among them. The one-way *r*ANOVA results of the STHT magnitudes among elastic seats at different frequencies also show that the STHT magnitudes are significantly different ($p < 0.05$) in the 4–5 Hz and 6.5–12 Hz frequency ranges for the NB condition, and in the 3–4.5 Hz, 8–12 Hz and 17.5–20 Hz for the WB condition (Table 3). The peak STHT magnitudes and corresponding frequencies for the elastic seats together with those of the rigid seat for both the sitting and three excitation conditions are summarized in Table 4. The mean peak magnitudes are obtained by averaging the peak

responses of individual participants that may occur at slightly different frequencies, while the corresponding frequency is obtained from the mean curves (Figure 6).

**Table 3.** *p*-values obtained from one-way *r*ANOVA of vertical STHT magnitudes between the elastic seats for two back support conditions and three levels of excitation.

| Frequency (Hz) | Without Back Support | | | With Back Support | | |
|---|---|---|---|---|---|---|
| | 0.25 m/s$^2$ | 0.50 m/s$^2$ | 0.75 m/s$^2$ | 0.25 m/s$^2$ | 0.50 m/s$^2$ | 0.75 m/s$^2$ |
| 2 | 0.375 | 0.258 | 0.198 | 0.074 | 0.065 | 0.055 |
| 3 | 0.183 | 0.056 | <0.05 | <0.05 | <0.05 | <0.01 |
| 4 | <0.001 | <0.001 | <0.001 | <0.001 | <0.001 | <0.001 |
| 4.5 | <0.001 | <0.001 | <0.001 | <0.01 | <0.01 | <0.05 |
| 5 | <0.01 | <0.001 | <0.05 | <0.001 | <0.05 | 0.333 |
| 5.5 | <0.05 | <0.01 | 0.207 | <0.01 | 0.085 | 0.569 |
| 6 | 0.245 | 0.079 | <0.01 | 0.141 | 0.644 | 0.157 |
| 6.5 | <0.05 | <0.001 | <0.001 | 0.783 | 0.185 | <0.05 |
| 7 | <0.05 | <0.001 | <0.001 | 0.592 | <0.05 | <0.01 |
| 7.5 | <0.001 | <0.001 | <0.001 | 0.361 | <0.05 | <0.01 |
| 8 | <0.001 | <0.001 | <0.001 | <0.05 | <0.05 | <0.001 |
| 8.5 | <0.001 | <0.001 | <0.001 | <0.05 | <0.05 | <0.001 |
| 9 | <0.001 | <0.05 | <0.001 | <0.05 | <0.05 | <0.01 |
| 10 | <0.001 | <0.01 | <0.001 | <0.05 | <0.05 | <0.01 |
| 12 | <0.001 | <0.05 | <0.05 | <0.01 | <0.05 | <0.001 |
| 15 | <0.01 | 0.251 | 0.397 | <0.05 | 0.538 | 0.083 |
| 17.5 | 0.140 | 0.066 | 0.272 | <0.001 | <0.001 | <0.05 |
| 20 | <0.01 | 0.069 | 0.541 | <0.001 | <0.001 | <0.001 |

**Table 4.** Mean (standard deviation) of the peak STHT magnitudes and the corresponding frequencies obtained for the 58 participants with two sitting conditions and three excitation levels.

| Excitation (m/s$^2$) | Seat A | | Seat B | | Seat C | | Rigid Seat | |
|---|---|---|---|---|---|---|---|---|
| | NB | WB | NB | WB | NB | WB | NB | WB |
| | Peak vertical STHT | | | | | | | |
| 0.25 | 2.39(0.36) | 2.09(0.26) | 2.62(0.37) | 2.20(0.23) | 2.57(0.41) | 2.10(0.24) | 2.17(0.34) | 1.89(0.18) |
| 0.50 | 2.31(0.33) | 2.04(0.27) | 2.59(0.35) | 2.14(0.20) | 2.52(0.39) | 2.08(0.30) | 2.10(0.27) | 1.86(0.16) |
| 0.75 | 2.33(0.33) | 2.07(0.27) | 2.65(0.39) | 2.14(0.23) | 2.50(0.44) | 2.09(0.28) | 2.16(0.27) | 1.88(0.19) |
| | Frequency corresponding to peak vertical STHT | | | | | | | |
| 0.25 | 5.36(0.61) | 5.14(0.59) | 5.25(0.54) | 5.08(0.55) | 5.32(0.53) | 5.05(0.67) | 5.42(0.66) | 5.06(0.57) |
| 0.50 | 4.95(0.49) | 4.75(0.51) | 4.88(0.42) | 4.70(0.41) | 4.91(0.46) | 4.63(0.52) | 4.99(0.52) | 4.72(0.52) |
| 0.75 | 4.68(0.44) | 4.53(0.39) | 4.54(0.40) | 4.44(0.36) | 4.65(0.45) | 4.38(0.45) | 4.82(0.42) | 4.52(0.43) |
| | Peak fore-aft STHT | | | | | | | |
| 0.25 | 2.86(0.73) | 2.38(0.51) | 3.00(0.85) | 2.33(0.55) | 2.93(0.77) | 2.22(0.55) | 2.58(0.62) | 2.56(0.34) |
| 0.50 | 2.63(0.79) | 2.17(0.44) | 2.67(0.74) | 2.11(0.44) | 2.56(0.72) | 1.98(0.41) | 2.32(0.55) | 2.28(0.39) |
| 0.75 | 2.56(0.83) | 2.07(0.45) | 2.64(0.72) | 2.02(0.48) | 2.40(0.74) | 1.88(0.44) | 2.27(0.56) | 2.12(0.32) |
| | Frequency corresponding to peak fore-aft STHT | | | | | | | |
| 0.25 | 5.02(0.80) | 4.88(0.77) | 4.86(0.66) | 4.50(0.65) | 4.91(0.91) | 4.80(1.06) | 5.19(0.81) | 4.81(0.61) |
| 0.50 | 4.69(0.60) | 4.58(0.59) | 4.69(0.51) | 4.49(0.64) | 4.84(0.65) | 4.55(0.81) | 4.91(0.67) | 4.58(0.65) |
| 0.75 | 4.62(0.55) | 4.45(0.52) | 4.53(0.46) | 4.34(0.54) | 4.62(0.60) | 4.42(0.70) | 4.70(0.59) | 4.38(0.53) |

NB: without back support; WB: with back support.

The results show that the peak vertical STHT magnitudes obtained for the elastic seats are higher than those obtained for the rigid seat, regardless of sitting and excitation conditions. The peak STHT magnitudes for seat B are higher than those obtained for other elastic seats (A and C). Furthermore, the peak vertical STHT magnitudes with seat C are higher than those with seat A. Slightly different trends are observed for the frequency corresponding to peak magnitudes for the two sitting conditions. For the NB sitting condition, the primary resonance frequencies are relatively higher for the rigid seat as

compared to the elastic seats for all the excitation conditions. Among the elastic seats, the primary resonance frequencies corresponding to peak vertical STHT are slightly higher for seat A compared with seats B and C, irrespective of sitting and excitation conditions. Moreover, the primary resonance frequencies for seat A are higher compared with the other elastic seats and the rigid seat for the WB sitting condition. The three-way *r*ANOVA results (Table 5) show that the elastic seat, the sitting condition and their interaction significantly influence the peak vertical STHT magnitude and the corresponding frequency.

**Table 5.** *p*–values obtained from a three-factor (S, BS and E) ANOVA on the primary resonance frequency and peak STHT magnitudes.

| Measure | S | BS | E | S × BS | S × E | BS × E | S × BS × E |
|---|---|---|---|---|---|---|---|
| Vertical resonance frequency | **<0.05** | **<0.001** | **<0.001** | **<0.05** | 0.537 | 0.870 | 0.477 |
| Vertical peak magnitude | **<0.001** | **<0.001** | 0.171 | **<0.001** | 0.556 | 0.527 | 0.097 |
| Fore-aft resonance frequency | 0.759 | **<0.05** | **<0.001** | 0.074 | 0.087 | 0.222 | 0.319 |
| Fore-aft peak magnitude | **<0.001** | **<0.001** | **<0.001** | 0.135 | 0.959 | 0.089 | 0.178 |

S-Seat (A, B, C), BS back support (NB and WB), E-excitation magnitude (0.25, 0.50 and 0.75 m/s$^2$ RMS acceleration).

The fore-aft STHT responses of the participants seated on the elastic and rigid seats for different sitting and excitation conditions are compared in Figure 7. Comparisons show nearly identical magnitudes at very low frequencies (up to 3 Hz) for all the elastic seats, while the most important effect is clearly evident in the peak magnitudes. The fore-aft STHT response magnitudes obtained with the rigid seat are lower than those obtained with the elastic seats in most of the frequency ranges, irrespective of the sitting and excitation conditions. Differences in the STHT magnitudes between the elastic and rigid seats are more pronounced near the primary resonance frequency magnitude and around 9 Hz.

The one-way *r*ANOVA of the fore-aft STHT magnitudes among elastic seats at different frequencies shows that the STHT magnitudes have statistically significant differences ($p < 0.05$) in broad ranges of frequencies (4–4.5 Hz and 10–20 Hz) for the NB sitting condition as compared with the WB sitting condition (4–5.5 Hz and 10–12 Hz) for all three excitation conditions (Table 6). Comparisons of the responses further show that differences in the peak STHT magnitudes between the elastic and rigid seats are greater for the NB condition as compared with the WB sitting condition for all the excitation conditions (Figure 7). Among the elastic seats, the peak magnitudes of both the PUF seats (seat A and B) are generally comparable and are slightly higher than those obtained with the air cushion (seat C).

The peak fore-aft STHT magnitudes and the corresponding frequencies for all the elastic seats, together with the rigid seat for two sitting and three excitation conditions, are shown in Table 4. The peak fore-aft STHT magnitudes observed with elastic seats are higher than those with the rigid seat for the NB sitting condition. An opposite trend, however, is observed for the WB sitting condition. Among the elastic seats, the peak magnitudes of seat B are relatively higher than those obtained for the other two seats (A and C) for the NB sitting condition, while the peak magnitudes of seat A are higher for the WB sitting condition. For the WB sitting condition, seat C yields lower peak magnitudes while the rigid seat yields higher magnitudes compared with other seats. Similar to the vertical STHT responses on the change in the primary resonance frequency, the effect is less for the elastic seats compared to the rigid seat. The three-way *r*ANOVA presented in Table 5 for the peak fore-aft STHT magnitudes and corresponding frequencies shows that the seat significantly affects the peak STHT magnitude ($p < 0.05$), while the primary resonance frequency is not significant.

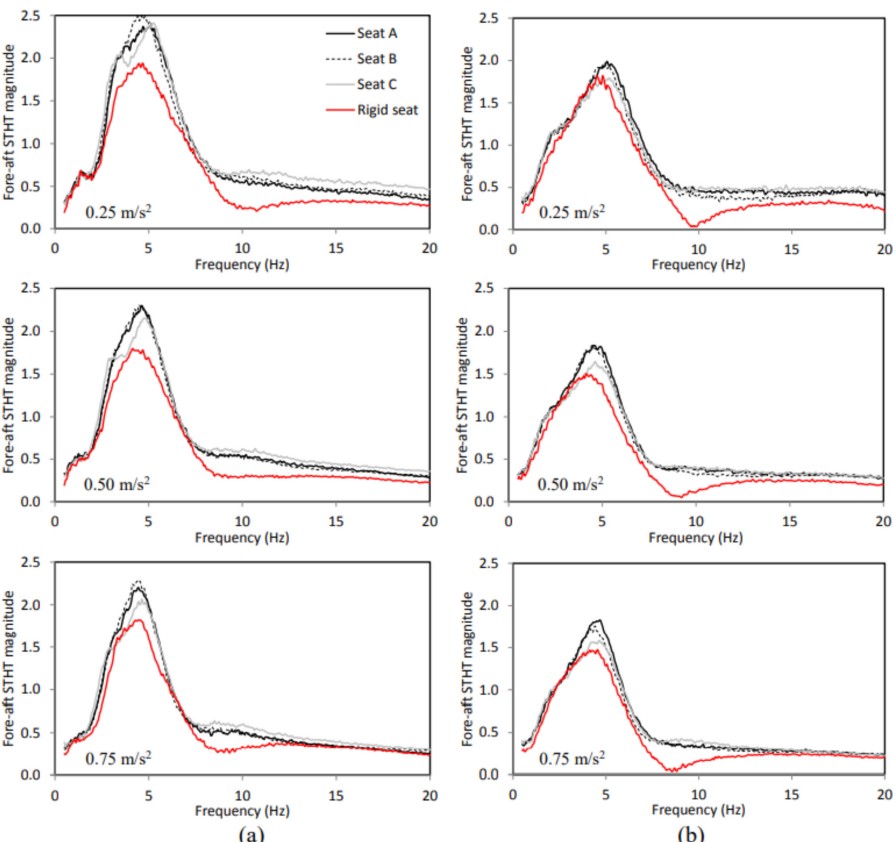

**Figure 7.** Mean fore-aft STHT magnitude of 58 participants sitting on the elastic and rigid seats and exposed to three excitations (0.25, 0.50 and 0.75 m/s² RMS acceleration): (**a**) without back support; and (**b**) with back support.

### 3.4. Effect of Back Support

Figure 8, as an example, compares the effect of back support on the vertical and fore-aft STHT responses under 0.50 m/s² RMS excitation for three elastic seats. The back support effect is also compared for the rigid seat under similar sitting and excitation conditions. The results show that mean vertical STHT responses with and without back support are comparable at lower frequencies (up to 3 Hz), irrespective of the seat. The STHT magnitudes obtained with the NB sitting condition are higher than those obtained with the WB sitting condition near the primary resonance frequency for all the seats. An opposite trend, however, is observed near the secondary resonance frequency (above 7 Hz). Similar trends are also observed for other excitation conditions. The peak STHT magnitudes with NB sitting conditions are relatively higher for the elastic seats B (18%) and C (17%) compared with the elastic seat A (12%) and the rigid seat (12%) for the peak STHT magnitudes with the WB condition (Table 4). Changing the sitting condition from the NB to the WB condition results in lower primary resonance frequencies for all the seats, irrespective of excitation conditions. The mean change in the primary resonance frequency with all three excitations is considerably more for seat C (0.27 Hz) compared with seats A (0.19 Hz) and B (0.15 Hz). A relatively higher reduction in the primary resonance frequency for the rigid seat is obtained (0.32 Hz) compared with the elastic seats when the sitting condition is changed from the NB to the WB condition. Table 5 shows that the peak vertical STHT magnitudes and the primary resonance frequencies with the NB and WB sitting conditions are significantly different ($p < 0.001$). Furthermore, the interactions between the seat and back support conditions are significantly different for the primary resonance frequency ($p < 0.05$) and peak vertical STHT magnitude ($p < 0.001$).

**Table 6.** *p*-values obtained from one-way *r*ANOVA of fore-aft STHT magnitudes among elastic seats for two sitting conditions and three levels of excitation.

| Frequency (Hz) | Without Back Support | | | With Back Support | | |
|---|---|---|---|---|---|---|
| | 0.25 m/s$^2$ | 0.50 m/s$^2$ | 0.75 m/s$^2$ | 0.25 m/s$^2$ | 0.50 m/s$^2$ | 0.75 m/s$^2$ |
| 2 | 0.586 | 0.312 | 0.131 | 0.887 | 0.922 | 0.119 |
| 3 | 0.567 | 0.331 | 0.198 | 0.474 | <0.05 | 0.089 |
| 3.5 | 0.670 | <0.01 | <0.001 | 0.638 | <0.01 | <0.05 |
| 4 | <0.001 | <0.001 | <0.01 | <0.05 | <0.001 | <0.01 |
| 4.5 | <0.01 | <0.01 | <0.01 | <0.05 | <0.001 | <0.001 |
| 5 | 0.489 | 0.295 | 0.634 | <0.01 | <0.001 | <0.01 |
| 5.5 | 0.146 | 0.647 | 0.635 | <0.01 | <0.05 | <0.01 |
| 6 | <0.01 | 0.170 | 0.059 | 0.067 | 0.204 | <0.01 |
| 6.5 | <0.01 | 0.266 | 0.074 | 0.137 | 0.460 | <0.01 |
| 7 | <0.01 | 0.127 | 0.574 | 0.221 | 0.273 | <0.05 |
| 7.5 | <0.05 | 0.565 | 0.261 | 0.058 | 0.987 | 0.755 |
| 8 | 0.179 | 0.496 | <0.05 | 0.939 | 0.877 | 0.104 |
| 8.5 | 0.181 | <0.05 | <0.001 | 0.695 | 0.310 | <0.05 |
| 9 | 0.177 | <0.05 | <0.001 | 0.683 | 0.102 | <0.05 |
| 9.5 | <0.01 | 0.194 | <0.001 | 0.060 | 0.091 | <0.01 |
| 10 | <0.01 | <0.001 | <0.001 | <0.05 | <0.01 | <0.01 |
| 10.5 | <0.01 | <0.01 | <0.001 | <0.05 | <0.01 | <0.01 |
| 11 | <0.01 | <0.001 | <0.001 | <0.05 | <0.05 | <0.05 |
| 11.5 | <0.001 | <0.01 | <0.01 | <0.05 | <0.05 | <0.01 |
| 12 | <0.001 | <0.01 | <0.001 | <0.05 | <0.05 | <0.01 |
| 12.5 | <0.001 | <0.001 | <0.001 | <0.05 | 0.064 | 0.063 |
| 15 | <0.001 | <0.001 | <0.01 | 0.169 | 0.094 | 0.096 |
| 17.5 | <0.001 | <0.01 | <0.01 | 0.417 | 0.506 | 0.668 |
| 20 | <0.001 | <0.001 | <0.01 | 0.667 | 0.900 | 0.794 |

The effect of back support on the mean fore-aft STHT response is more prominent than observed in the vertical STHT response, particularly near primary resonance. The peak STHT magnitudes of participants sitting with the NB condition are notably higher compared with the WB sitting condition, irrespective of the seat. A similar trend was also evident under other excitations. The mean reduction in the peak fore-aft STHT magnitudes considering all three excitation conditions is considerably high for the elastic seats, i.e., 18%, 22% and 23% for seats A, B and C, respectively, when the sitting condition is changed from NB to WB (Table 4). The peak magnitude is reduced by only 3% for the rigid seat when the sitting condition is changed from NB to WB. A change in the sitting condition from NB to WB results in lower primary resonance frequencies for all the seats, irrespective of excitation conditions. Similar to vertical STHT responses, the change in the primary resonance frequency is lower for the elastic seats (0.14, 0.25 and 0.20 Hz for seats A, B and C, respectively) as compared with the rigid seat (0.35 Hz) when the sitting condition is changed from NB to WB. Results in Table 5 show that the peak fore-aft STHT magnitude ($p < 0.001$) and the primary resonance frequency ($p < 0.05$) of the two sitting conditions are significantly different.

*3.5. Effect of Vibration Magnitude*

Figure 9 illustrates the mean vertical and fore-aft STHT responses obtained with the elastic seats together with the rigid seat under selected excitation magnitudes for the NB sitting condition. The results show a reduction in the primary resonance frequency of the vertical STHT responses with an increase in excitation magnitude for all the seats. The softening tendency is more pronounced with an increase in excitation from 0.25–0.50 m/s$^2$ RMS, while this effect is smaller with a further increase in excitation from 0.50–0.75 m/s$^2$ RMS. A similar softening tendency of the human body is also obtained for the WB sitting condition. Reduction in the primary resonance frequency is relatively more for the NB compared with the WB condition for the elastic and rigid seats. Such changes in frequency,

however, are relatively small for the rigid seat. Increasing the excitation level from 0.25 to 0.75 m/s² RMS results in comparable reductions in the primary resonance frequency with the elastic seats. These were obtained as 0.68, 0.71 and 0.67 Hz for the NB sitting condition, and 0.63, 0.64 and 0.67 Hz for the WB sitting condition for seats A, B and C, respectively (Table 4). The corresponding reductions for the rigid seat were obtained as 0.60 and 0.54 Hz, respectively, for the NB and WB sitting conditions. The peak vertical STHT magnitudes obtained for different excitations, however, are comparable for the elastic and rigid seats (Figure 9). Results obtained from three-way *r*ANOVA suggest that the primary resonance frequency significantly ($p < 0.001$) decreases with an increase in excitation magnitude; however, the peak magnitude is not significant (Table 5).

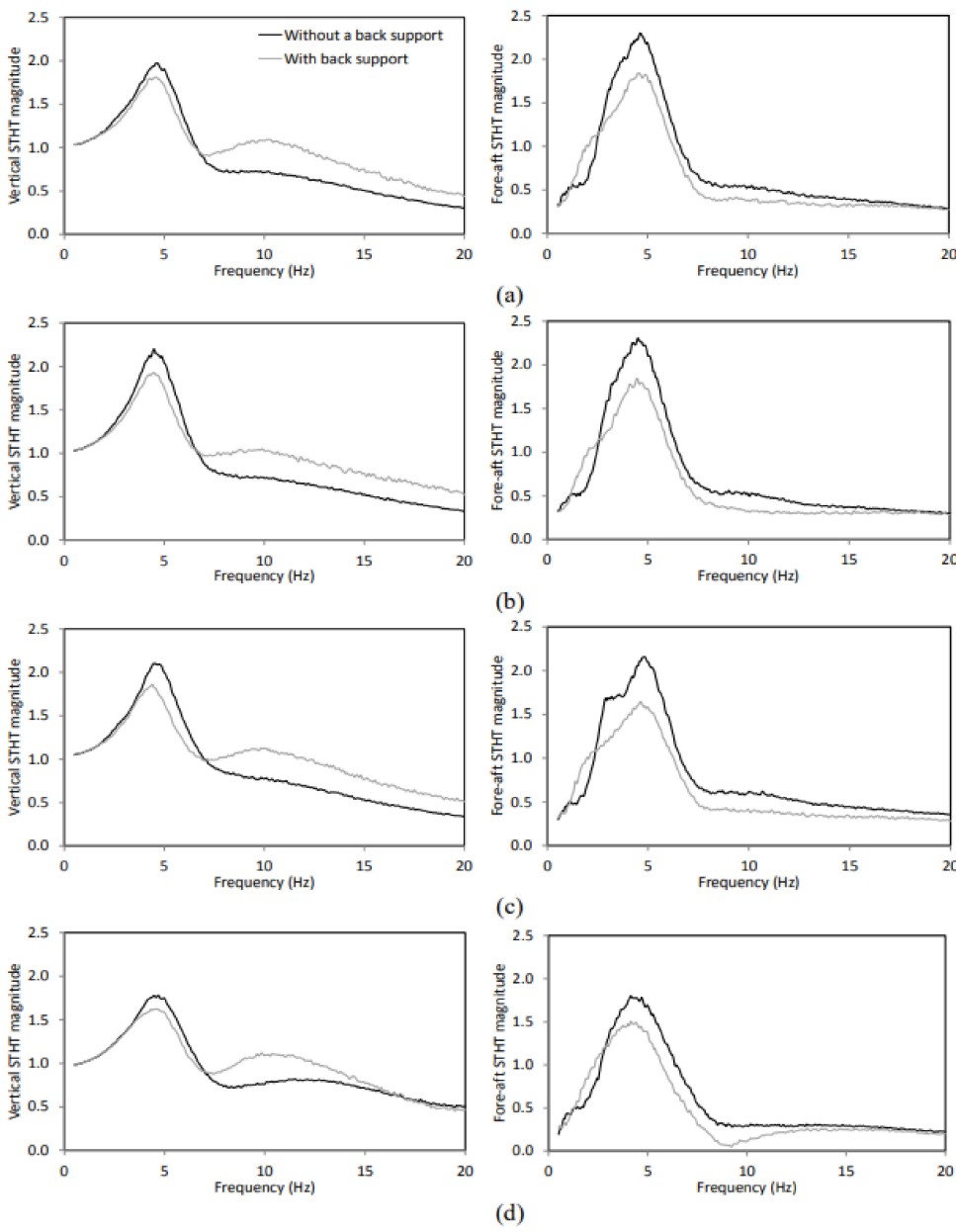

**Figure 8.** Mean STHT magnitudes of 58 participants seated with and without back support under 0.50 m/s² RMS excitation: (**a**) seat A; (**b**) seat B; (**c**) seat C; and (**d**) rigid seat.

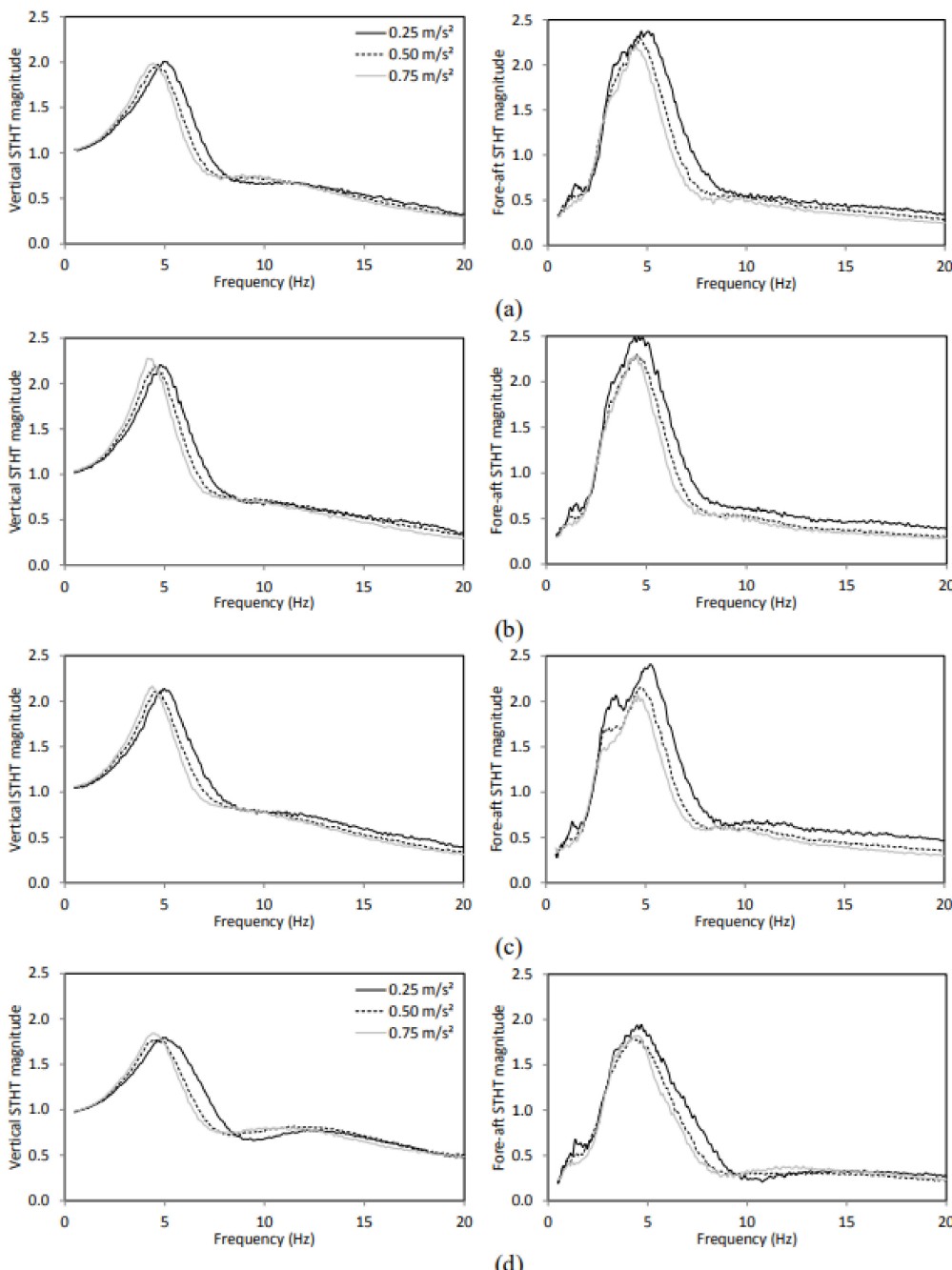

**Figure 9.** Influence of excitation magnitude on the mean STHT magnitude responses of 58 participants sitting without back support: (**a**) seat A; (**b**) seat B; (**c**) seat C; and (**d**) rigid seat.

Unlike the vertical STHT, the fore-aft STHT responses exhibit notable reductions in peak magnitude with an increase in excitation for all the seats (Figure 9). Reduction in the peak magnitude is more for seat C (18%) compared with seats A (10%) and B (12%), with an increase in excitation from 0.25–0.75 m/s$^2$ RMS for the NB sitting condition. Reduction in the peak magnitude is comparable for the elastic seats (13%, 13% and 15% for seats A, B and C, respectively), with the increase in excitation for the WB sitting condition (Table 4). Reduction in the peak fore-aft magnitude for the rigid seat is slightly lower for the NB sitting condition (14%) than the WB sitting condition (17%). Like vertical STHT, there is a softening tendency of the human body with an increase in excitation for the fore-aft STHT responses; however, it is not clearly evident in the fore-aft responses (Figure 9). The softening tendency calculated from the primary resonance frequency in Table 4 shows that

elastic seats and back support conditions are coupled with the softening tendency. For the NB sitting condition, a reduction in the primary resonance frequency is relatively more for seat A (0.40 Hz) compared with seat B (0.33 Hz) and seat C (0.29 Hz), with an increase in excitation from 0.25–0.75 m/s$^2$ RMS. The softening tendency is more for seats A (0.43 Hz) and C (0.38 Hz) compared with seat B (0.16 Hz) for the WB support with an identical increase in excitation. For the rigid seat, the softening tendency is slightly higher for the NB sitting condition (0.49 Hz) compared with the WB support (0.44 Hz). Three-way *r*ANOVA shows that the primary resonance frequency and the peak magnitude are significantly ($p < 0.001$) different with an increase in excitation (Table 5).

### 3.6. Correlations with Gender and Anthropometric Parameters

As an example, Tables 7 and 8 show the regression coefficient values between the anthropometric parameters (mass-, stature- and build-related measures) and the peak vertical and fore-aft STHT magnitudes, respectively, for three excitations and two sitting conditions for seat A. The table also shows the regression values for the primary resonance frequencies. The results show that there is no correlation between the peak magnitudes of both the STHT responses and any of the anthropometric parameters, for both genders, irrespective of the sitting condition and vibration excitation. For the vertical responses, $r^2$ values with the primary resonance frequency of the vertical STHT responses are relatively higher ($r^2 = 0.11$–$0.23$) with the hip circumference, body fat mass and percent body fat for the NB sitting condition than those for the WB sitting condition. The $r^2$ values are more than 0.1 between the peak fore-aft STHT magnitudes and the lean body mass for the NB sitting condition. For the NB sitting condition, the $r^2$ values are more than 0.1 between the primary resonance frequency and the seat pan contact area.

**Table 7.** Values of $r^2$ between anthropometric parameters and the peak vertical STHT magnitude, as well as primary resonance frequency, for three excitations and two sitting conditions for seat A.

| Anthropometric Parameters | | Primary Resonance Frequency | | | | | | Peak Magnitude | | | | | |
|---|---|---|---|---|---|---|---|---|---|---|---|---|---|
| | | Without Back Support | | | With Back Support | | | Without Back Support | | | With Back Support | | |
| | | 0.25 m/s$^2$ | 0.50 m/s$^2$ | 0.75 m/s$^2$ | 0.25 m/s$^2$ | 0.50 m/s$^2$ | 0.75 m/s$^2$ | 0.25 m/s$^2$ | 0.50 m/s$^2$ | 0.75 m/s$^2$ | 0.25 m/s$^2$ | 0.50 m/s$^2$ | 0.75 m/s$^2$ |
| Mass-related | BMI | 0.047 | 0.038 | 0.061 | 0.003 | 0.016 | 0.001 | 0.024 | 0.001 | 0.002 | 0.002 | 0.035 | 0.014 |
| | Body fat mass | 0.170 | 0.120 | 0.162 | 0.092 | 0.111 | 0.035 | 0.011 | 0.011 | 0.007 | 0.011 | 0.047 | 0.009 |
| | Percent body fat | 0.231 | 0.113 | 0.113 | 0.294 | 0.239 | 0.108 | 0.001 | 0.007 | 0.004 | 0.048 | 0.058 | 0.007 |
| | Lean body mass | 0.002 | 0.003 | 0.011 | 0.073 | 0.017 | 0.017 | 0.041 | 0.002 | 0.001 | 0.032 | 0.008 | 0.001 |
| Build-related | Hip circumference | 0.181 | 0.167 | 0.161 | 0.053 | 0.127 | 0.039 | 0.034 | 0.037 | 0.039 | 0.004 | 0.009 | 0.001 |
| | Seat-pan contact area | 0.022 | 0.062 | 0.056 | 0.005 | 0.072 | 0.030 | 0.001 | 0.001 | 0.001 | 0.008 | 0.001 | 0.001 |
| | Mean contact pressure | 0.065 | 0.034 | 0.017 | 0.075 | 0.056 | 0.020 | 0.014 | 0.011 | 0.024 | 0.016 | 0.009 | 0.001 |
| Stature-related | Stature | 0.001 | 0.009 | 0.019 | 0.060 | 0.011 | 0.003 | 0.029 | 0.014 | 0.026 | 0.068 | 0.042 | 0.012 |
| | Sitting height | 0.002 | 0.001 | 0.001 | 0.036 | 0.022 | 0.026 | 0.050 | 0.020 | 0.017 | 0.044 | 0.043 | 0.010 |
| | C7 height | 0.007 | 0.005 | 0.004 | 0.057 | 0.006 | 0.013 | 0.135 | 0.047 | 0.047 | 0.168 | 0.066 | 0.054 |

**Table 8.** Values of $r^2$ between anthropometric parameters and the peak fore-aft STHT magnitude, as well as primary resonance frequency, for the three excitations and the two sitting conditions for seat A.

| Anthropometric parameters | | Primary Resonance Frequency | | | | | | Peak Magnitude | | | | | |
|---|---|---|---|---|---|---|---|---|---|---|---|---|---|
| | | Without Back Support | | | With Back Support | | | Without Back Support | | | With Back Support | | |
| | | 0.25 m/s² | 0.50 m/s² | 0.75 m/s² | 0.25 m/s² | 0.50 m/s² | 0.75 m/s² | 0.25 m/s² | 0.50 m/s² | 0.75 m/s² | 0.25 m/s² | 0.50 m/s² | 0.75 m/s² |
| Mass-related | BMI | 0.017 | 0.031 | 0.027 | 0.027 | 0.036 | 0.006 | 0.063 | 0.056 | 0.080 | 0.003 | 0.005 | 0.003 |
| | Body fat mass | 0.065 | 0.048 | 0.046 | 0.037 | 0.090 | 0.040 | 0.011 | 0.008 | 0.022 | 0.001 | 0.015 | 0.002 |
| | Percent body fat | 0.079 | 0.055 | 0.047 | 0.025 | 0.052 | 0.061 | 0.034 | 0.063 | 0.039 | 0.004 | 0.005 | 0.009 |
| | Lean body mass | 0.006 | 0.001 | 0.001 | 0.019 | 0.004 | 0.004 | 0.115 | 0.193 | 0.191 | 0.014 | 0.014 | 0.016 |
| Build-related | Hip circumference | 0.081 | 0.031 | 0.040 | 0.058 | 0.074 | 0.041 | 0.016 | 0.054 | 0.071 | 0.004 | 0.033 | 0.004 |
| | Seat-pan contact area | 0.113 | 0.124 | 0.125 | 0.002 | 0.059 | 0.018 | 0.003 | 0.027 | 0.034 | 0.001 | 0.001 | 0.018 |
| | Mean contact pressure | 0.032 | 0.082 | 0.062 | 0.032 | 0.022 | 0.007 | 0.019 | 0.051 | 0.018 | 0.004 | 0.029 | 0.022 |
| Stature-related | Stature | 0.006 | 0.004 | 0.001 | 0.027 | 0.006 | 0.001 | 0.065 | 0.185 | 0.155 | 0.019 | 0.046 | 0.071 |
| | Sitting height | 0.001 | 0.008 | 0.010 | 0.006 | 0.002 | 0.001 | 0.011 | 0.051 | 0.062 | 0.001 | 0.001 | 0.028 |
| | C7 height | 0.002 | 0.001 | 0.002 | 0.003 | 0.002 | 0.001 | 0.050 | 0.083 | 0.090 | 0.004 | 0.043 | 0.057 |

## 4. Discussion

Vibration transmission characteristics of the elastic seats are nonlinear and depend on supported body mass (Figure 3). Therefore, comparable driving point excitation is important for comparing the STHT responses of the body seated on the three elastic seats of different characteristics. In the present study, excitation magnitude was controlled at the body-seat interface for all the elastic seats while considering the effects of vibration excitation and the body mass of participants. Therefore, this study is considered to provide a meaningful comparison of the STHT responses among the elastic seats. Furthermore, the coupling characteristics of the body on the elastic seats are different from those on the rigid seat. The body-seat contact area is considerably higher, while the mean contact pressure is considerably lower on the elastic seats compared with the rigid seat (Figure 2; Table 1). Thus, it is important to compare the responses obtained on the elastic seats with those on the rigid seat.

### 4.1. STHT Response Characteristics

The STHT responses of the participants seated on the three elastic seats show large inter-subject variations in terms of magnitude (Figures 4 and 5). The peak CoV of the vertical STHT magnitude near the primary resonance frequency is slightly higher (35%, 30% and 34% for seats A, B and C, respectively, with the NB sitting condition) than the 29% CoV obtained on the rigid seat [7]. For fore-aft STHT magnitude responses, the peak CoVs obtained for seat A (46%) and seat C (44%) are comparable with the rigid seat (44%), but they are slightly lower for seat B (39%). The scatter of the STHT magnitude in the present study is considerably higher than those reported in Wang et al. [11] and M-Pranesh et al. [8] for vertical STHT magnitudes (15–20%). The scatter in the fore-aft STHT magnitudes of the present study is also higher, 30–40%, compared with the results reported by Wang et al. [11] and M-Pranesh et al. [8]. Synthesis of the vertical STHT responses in Rakheja et al. [10] also reported relatively lower peak CoV in the magnitude data, of about 29% near 7 Hz. Large scatter in the STHT magnitude responses have also been obtained in previous studies, but CoV values were not reported [1,13,33,34]. Lower variability while sitting with the WB condition compared with the NB condition in the present study is in line with the results of previous studies [7,11,34]. Scatter in the STHT responses may be due to the different characteristics of the participants [34], but higher variability in the present study could also be due to different characteristics of the elastic seats.

The results of the vertical STHT responses show a slight variation in the primary resonance frequency of the body seated on seats A (4.00–6.38 Hz), B (4.06–5.63 Hz) and C (4.06–6.19 Hz). This range of the primary resonance frequency is comparable to that of the rigid seat response (4.13–6.00 Hz for the vertical STHT responses), as reported by Dewangan et al. [7], and it also aligns with that of the foam seat response (4.7 Hz) reported in Kim et al. [4]. Boileau et al. [35] and Rakheja et al. [10] reported that the majority of the datasets in the STHT indicate a dominant peak occurring within 4–6 Hz. The primary resonance frequency for the fore-aft STHT responses of the present study (3.50–6.00, 3.31–5.81 and 3.31–5.94 Hz for seats A, B and C, respectively, for the NB sitting condition) is relatively lower than 3.88–6.31 Hz, as reported in Dewangan et al. [7] with the rigid seat for the NB sitting condition. Results show a peak in the 1–2 Hz range for the fore-aft magnitude responses of most of the participants seated with the NB sitting condition (Figure 5). A similar trend has been reported for the fore-aft responses of participants seated on the rigid seat [7].

Dewangan et al. [7] suggested that variations in the STHT magnitude and primary resonance frequency measured with the rigid seat are due to gender and anthropometric characteristics of the participants, as well as the sitting and excitation conditions. Relatively higher CoVs of the STHT magnitude and primary resonance frequency with the elastic seats compared to the rigid seat suggested likely contributions due to body coupling with the elastic seats. It should be noted that rigid seat results reported by Dewangan et al. [7] were obtained with the same participants of the present study. Furthermore, variation in the responses among the three elastic seats may also be due to similar reasons; i.e., the visco-elastic properties and contouring of the seats changed the body seat contact area, mean contact pressure and pelvis orientation, in addition to pelvic rotation. Pelvic motion is influenced due to both pelvic orientation and vibration frequency [36]. Pelvic rotation and femur displacement are more significant with the softer cushion compared with the stiff cushion [25]. Sitting on the air cushion (seat C) resulted in a more uniform body-seat pressure distribution compared with the PUF (seats A and B). The air bubbles in seat C are connected together. The localized contact pressure changes the airflow from one bubble to another until equilibrium pressure distribution is attained, and thus, more uniform contact pressure was observed in the present study. More uniform contact pressure has also been reported on the air cushion compared with the PUF seats [18,24]. The PUF had a wider inter-subject variability in terms of the peak ischial pressure, which indicates that pressure-redistributing characteristics of the PUF seat may vary with the subject characteristics [24]. Akgunduz et al. [19] reported that body size and mass have a definite effect on the contact area and contact pressure at the body-seat interface on the PUF seats, while sitting posture and cushion characteristics have the most significant effects on the contact area and contact pressure. Thus, the coupling of the body on different characteristics of the elastic seats has caused different degrees of the pelvis orientation and pelvis rotation of the seated body, and likely variations in the STHT responses with different seats. Since the effect on the vertical and fore-aft responses of all the participants cannot be compared from individual responses, the mean vertical and STHT responses of all the participants would be appropriate by which to compare the elastic seats' responses, and also with the responses obtained on the rigid seat.

### 4.2. Comparisons of Mean Responses

This study revealed higher peak vertical STHT magnitudes for the participants seated on the elastic seats than those on the rigid seat, regardless of sitting and excitation conditions (Figure 6, Table 4). Hinz et al. [29] reported a higher peak magnitude for the suspended seat compared with the rigid seat. Furthermore, the magnitude near the resonance is broader and shows two or three local peaks with the suspended seat. The higher vertical STHT magnitude for the elastic seats compared with the rigid seat may be due to the pelvis motion. Beck et al. [25] reported higher pelvic rotation with a softer cushion compared with a stiffer cushion. Zimmermann and Cook [36] reported considerably more pelvis motion at frequencies of vibration up to 6 Hz than at frequencies greater than 6 Hz. The study further opined that the anterior-posterior motion of the pelvis contributes to the

peak STHT magnitude in the 4.5–6 Hz range. According to Zimmermann and Cook [36] and Beck et al. [25], a soft seat may exhibit a higher peak compared with a stiff seat. This study presented the variations in the peak vertical STHT magnitudes among the elastic seats (Figure 6, Table 4). Vibration transmissibility characteristics of both PUFs (seats A and B) are comparable (Figure 3), although the relatively stiff and contoured PUF (seat B) showed a higher peak magnitude compared with the flat and less stiff PUF (seat A). This indicates that, in addition to pelvis rotation, body-seat contact area and mean contact pressure influence peak magnitude. Kitazaki and Griffin [15] reported that the fourth mode at 4.9 Hz, which corresponds to the peak magnitude, is due to the vertical motion of the head, spinal column and pelvis due to axial and shear deformations of the buttocks tissue, in phase with a vertical visceral mode, and a bending mode of the upper thoracic and the cervical spine. A higher body-seat contact area causes more body surface exposure to vibration, while lower mean contact pressure may cause lower buttock tissue compression. The flat PUF (seat A) shows a slightly higher contact area and lower mean contact pressure (Figure 2) compared with the air cushion (seat C). Furthermore, seat A is relatively soft and damped compared with seat C. These may have contributed to a slightly higher peak STHT magnitude with seat C compared with seat A. The considerably lower peak STHT magnitude for the rigid seat compared with the elastic seats may also result from the lower contact area, high mean peak pressure, pelvis orientation and rotation of the seated body on the rigid seat (Figure 2; Table 1). Higher peak vertical STHT magnitudes with elastic seats in the present study and for the suspension seat [29] compared with the rigid seat, however, suggest a trend opposite to to-the-body biodynamic responses measured in terms of apparent mass (AM) on the elastic and rigid seats [22,23,37].

The vertical STHT responses of relatively fewer participants showed a secondary resonance peak for the NB sitting condition on the elastic seats, and thus, the peak near the secondary resonance is not evident in the mean responses (Figure 6). However, a distinct secondary peak for many participants with the rigid seat was observed; thus, a clear peak near the secondary resonance on the mean vertical STHT responses was evident [7]. This may partly be the reason for the lower STHT magnitude near secondary resonance for the NB sitting condition for the elastic seats compared with the rigid seat (Figure 6). Dewangan et al. [22,23] reported that AM magnitude near secondary resonance with the elastic seats was generally lower than the rigid seat. The vertical STHT magnitude near secondary resonance is comparable between the elastic and rigid seats for the WB sitting condition (Figure 6). A backrest serves as a constraint for the pitch motion of the pelvis and bending motion of the spine in the sagittal plane, and thus, influences the transmission of vibration to the head. Back support limits the transmission of vibration to the head for all the seats, which may be one of the reasons for the comparable STHT magnitude of the participants for the WB sitting condition for the elastic and rigid seats.

Sandover and Dupuis [16] suggested that the resonances of the human body are related to bending in the lumbar spine caused by a rocking motion of the pelvis. Variation in the primary resonance frequency of the STHT responses in different seats may be due to contributory factors, such as the stiffness of seats, vibration transmissibility characteristics of seats, contact area, mean contact pressure and sitting conditions for the bending motion of the spine and rocking motion of the pelvis. The vertical STHT responses of the seats show lower primary resonance frequency for the elastic seat B than those obtained for the other elastic seats (A and C) for the NB sitting condition (Table 4). The rigid seat shows higher primary resonance frequency for the vertical STHT responses compared with the elastic seats. The resonance frequency of the elastic seats is comparable (Figure 3d), and thus, variation in the primary resonance frequency among various seats may be due to other contributory factors. Lemerle and Boulanger [26] observed no significant effect of the lower limb motion on the primary resonance frequency of the human body on the AM responses. Payne and Band [38] hypothesized that increasing the body-seat contact area decreases the total axial stiffness under the pelvis due to the non-linear force-defection relationship of the tissue. Owing to higher mean contact pressure on the rigid seat, the buttock tissues are stiff and

cause higher resonance frequency. On the other hand, the lower mean contact pressure on elastic seat B (Figure 2) caused low stiffness of the buttock tissue and lower primary resonance frequency compared with the other elastic seats (seats A and C). Wu et al. [20,21] observed that the contact pressure distribution on the body-seat interface is the contributory factor for the variation in the resonance frequency of the seat under vertical vibration. The contoured PUF (seat B) causes a considerably higher contact area (Figure 2), while the rigid seat causes a considerably lower contact area (Table 1). It may also be partly the reason for the lower primary resonance frequency with seat B and higher primary resonance frequency with the rigid seat for the NB sitting condition. Sitting with a vertical back support changes muscle stiffness, and thus, might also have affected the primary resonance frequency.

The fore-aft motion of the head is a combined effect of the deformation of buttock tissues, pitching motion of the pelvis, bending motion of the spine and pitching of the head [15]. Relatively higher magnitudes of the fore-aft STHT responses compared with the vertical STHT responses are evident, although participants are exposed to vertical excitation on the elastic seats (Table 4; Figure 7). This trend is in line with the results of previous studies conducted on a rigid seat [8,11]. Furthermore, variation in the fore-aft STHT responses may again be due to the coupled effects of the body-seat contact area, mean contact pressure, static stiffness, vibration transmissibility and sitting conditions. Most of the subject's fore-aft STHT responses for the NB sitting condition on the elastic seats showed a peak in the range of 1.5–2.0 Hz, which is in line with the results of the rigid seat [7,11]. The secondary resonance peak obtained in the frequency range of 9–14 Hz in the rigid seat [7] is not clearly evident on the elastic seats.

### 4.3. Effect of Back Support

The peak vertical STHT magnitudes significantly reduced ($p < 0.001$) when the sitting condition was changed from the NB to the WB sitting condition for the elastic seats (Table 5). Studies on the vertical STHT responses have also reported a similar effect of peak magnitude on the elastic seats [29] and rigid seat [7,8,11,12]. The variation in the STHT magnitude between the NB and WB sitting conditions may be attributed to sitting posture. Seated posture changes the demands on the musculoskeletal system because of a tendency of the body to adopt a position in which the pelvis is rotated backward to compensate for muscular tightness in the hamstring muscles [39]. Andersson [40] reported that the shape of the lumbar spine during sitting, to some degree, is controlled by the rotation of the pelvis. In the absence of a backrest, the lumbar spine tends to assume a lordotic posture (inward curvature of the lumbar spine) to obtain balance [41]. Back support causes an increase in lumbar lordosis and derotation of the pelvis [40]. Thus, the pelvic orientation between sitting with and without vertical back support is different. A backrest also serves as a constraint for the motion of the spine and may reduce axial deformation of the lower lumbar spine, influencing the transmission of vibrations to the head. Furthermore, the change in a sitting condition also changed body-seat contact area and mean contact pressure (Figure 2). Dewangan et al. [23] reported a decrease in the contact area and mean contact pressure on elastic and rigid seats when the sitting condition was changed from the NB to the WB sitting condition.

The peak vertical STHT magnitudes varied among different seats when the sitting condition was changed from the NB to the WB sitting condition. Reduction in the peak STHT is considerably large for elastic seats B (18%) and C (17%) compared with seat A (12%). The effect of changes in the sitting position on the peak vertical STHT magnitude is also relatively less on the rigid seat (12%). Relatively higher peak STHT magnitude for seats B and C were recorded compared with elastic seat A and the rigid seat for the NB condition, which might be the reason for a considerable reduction in the peak magnitude when the sitting condition is changed from the NB to the WB sitting condition. The variation in the peak STHT magnitude among the elastic and rigid seats may also be due to variations in the characteristics of the seats, pelvis orientation and pelvis rotation. In the present study, the elastic seat cushion mounted on the seat structure was subjected to compression and relaxation under vibration excitation, while the backrest was stationary relative to the

seat structure. There is relative motion between the body-seat interface at the seat pan and the backrest, and thus, friction between the back of the participants and the seat. A backrest stabilizes the posture by relieving the amount of effort required to fight gravity, while also being an additional constraint to the motion of the spine. Furthermore, a relative motion may depend on the damping properties of the seat and vibration transmissibility characteristics of the elastic seats.

The results show significantly lower primary resonance frequency ($p < 0.001$) of the participants seated with vertical back support than without back support (Tables 4 and 5). A similar trend was reported by Hinz et al. [29] and Wang et al. [11]. Reduction in the primary resonance frequency when the sitting condition is changed from NB to WB is due to a change in muscle tension. Muscles are more active, and thus, become stiff when humans sit without back support; however, back support reduces muscle load. Thus, sitting without back support causes higher resonance frequency than those obtained with back support.

The peak fore-aft STHT magnitudes significantly reduced ($p < 0.001$) when the sitting condition was changed from the NB to the WB sitting condition for the elastic seats (Table 5). Studies on the fore-aft STHT responses have also reported a reduction in the peak magnitude on the rigid seat [7,8,11]. Furthermore, the effect of back support on the peak magnitude is more on the fore-aft STHT responses compared with the vertical STHT response, which is in line with the results in Wang et al. [11]. A backrest helps limit horizontal and rotational body motions, particularly in the sagittal plane, and thereby alters the fore-aft vibration of the head; thus, the effect on the fore-aft response is more.

*4.4. Effect of Vibration Magnitude*

The vertical and fore-aft STHT responses of the participants seated on the elastic and rigid seats show a significant ($p < 0.001$) decrease in the primary resonance frequency with an increase in vibration excitation (Figure 9, Tables 4 and 5). Measurements of the STHT responses of human participants seated on a rigid seat and exposed to vertical whole-body vibration have been reported a softening tendency in the body [7,8,11,13,42]. The elastic seats (0.68, 0.71 and 0.67 Hz for seats A, B and C, respectively, for the NB condition, and 0.61, 0.64 and 0.67 Hz for seats A, B and C, respectively, for the WB condition) show a comparable softening tendency of the body for the vertical STHT responses; however, it is relatively more compared with the rigid seat (0.60 and 0.53 Hz for the NB and WB sitting conditions, respectively), when the excitation magnitude was increased from 0.25–0.75 m/s$^2$ RMS (Table 4). For the fore-aft STHT responses, they had slightly different values of softening tendency among the elastic seats (0.40, 0.33 and 0.29 Hz for seats A, B and C, respectively, for the NB condition and 0.43, 0.16 and 0.38 Hz for seats A, B and C, respectively, for the WB condition) and were lower than the rigid seat (0.49 and 0.44 Hz for the NB and WB conditions, respectively). Back support showed a slightly lower softening effect in the present study, which is in line with the results in Wang et al. [11]; M-Pranesh et al. [8].

The present study showed a comparable vertical peak and significantly low peak fore-aft STHT magnitude due to changes in vibration magnitude (Tables 4 and 5), which is in line with the reported studies [7,8,42]. Matsumoto and Griffin [42] reported that the frequency of the peak and the corresponding transmissibility of pitch motion of the head decreases with the increase in the vibration magnitude, which may be the reason for the low peak magnitude for the fore-aft response with the increase in excitation magnitude.

## 5. Conclusions

The seat-to-head vibration transmission characteristics of 58 participants seated on three different elastic seats were investigated under three different levels of vertical vibration and two sitting conditions. The STHT responses were strongly influenced by the body-seat contact area, mean contact pressure and seat stiffness, which affected pelvis orientation and pelvis rotation. The STHT responses of the body coupled with the elastic seats considerably differed from those with a rigid seat. Furthermore, the vertical and fore-aft STHT responses were strongly coupled with sitting conditions and excitation

magnitude in a complex manner. The peak vertical and fore-aft STHT magnitudes were significantly ($p < 0.001$) different among the elastic seats, and were generally higher than those obtained with the rigid seat. Peak vertical magnitude responses of relatively stiff and contoured PUF (seat B) were higher compared with other elastic seats, irrespective of sitting and excitation conditions. The primary resonance frequencies of the vertical STHT responses were also significantly ($p < 0.05$) different among elastic seats, and were higher for the soft and flat PUF (seat A) compared with other elastics seats, irrespective of sitting conditions. Changes in the back support conditions significantly alter the peak magnitudes and primary resonance frequencies. The effect of sitting conditions on the vertical STHT responses was considerably more for seats A (flat PUF) and C (air cushion) compared with seat B (contoured PUF). An increase in excitation magnitude significantly ($p < 0.001$) decreased the primary resonance frequency; however, the effect of excitation was comparable to vertical STHT responses for all the seats, irrespective of sitting conditions. Furthermore, an increase in excitation significantly ($p < 0.001$) decreased the peak fore-aft magnitude and the effect was more on seat C compared with seats A and B. The vibration biodynamic responses of the seated body, as described in the international standard [43], are based solely on responses obtained with rigid seats. The results obtained in this study suggest the need for further studies on the effects of coupling with elastic seats in order to establish more reliable vibration biodynamic responses for the design and assessment of seats.

**Author Contributions:** K.N.D., Conceptualization, methodology, software and formal analysis, data curation, writing—original draft preparation; Y.Y., validation, investigation, resources, funding acquisition, writing—review and editing; S.R., supervision, project administration, funding acquisition. All authors have read and agreed to the published version of the manuscript.

**Funding:** This research was funded by [National Natural Science Foundation of China] grant number [52105114] and [National Foreign Expert Program] grant number [G2022013006].

**Data Availability Statement:** Not applicable.

**Acknowledgments:** The authors are grateful for the funding support from the National Natural Science Foundation of China (Project No.: 52105114) and the National Foreign Expert Program (Project No.: G2022013006).

**Conflicts of Interest:** The authors declare no conflict of interest.

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
