# Peer review of "Seat-to-Head Transmissibility Responses of Seated Human Body Coupled with Visco-Elastic Seats"

_vibration, doi:10.3390/vibration5040051_

Round 1
Reviewer 1 Report
1) The vibration frequencies within the studied range may coincide with the frequencies of the rhythms of the cerebrum. It seems to be necessary to investigate the influence of these frequencies.
2) The authors must provide DOI indexes to all references, if exist.
Author Response
Thanks for your review!
Please check the attached file.

Reviewer 2 Report
Main comments
The paper presents a study of the transmission of vibration through seats and through the human body. The effect of various factors has been investigated including the presence of a back support and vibration magnitude. A clear and well-presented paper. I offer my comments on the manuscript.
The line numbers below correspond to those on the draft manuscript.
Comments
Line 35
Change “… Padan …” to “… Paddan …”. Reference ‘Paddan and Griffin 1988’ does not appear in the references list.
Line 62
Change “… human have …” to “… humans have …”. Change “… with body …” to “… with the body …”.
Line 76
Change “… compared of the …” to “… compared the …”.
Line 79
Change “… Dewangan 2015 …” to “… Dewangan et al., 2015 …”.
Line 101
Change “… Both the studies …” to “… Both studies …”.
Line 118
The vibration magnitude should be presented as ‘m/s2 r.m.s.’. This applies throughout the manuscript.
Line 153
Change “… M-Pranesh (2010) …” to “… M-Pranesh et al. (2010) …”.
Line 208
Change “… participated the …” to “… participated in the …”.
Line 284-5
Change “… correspond to …” to “… corresponding to …”.
Line 297
Label ‘c’ is missing from the figure.
Line 421
This section discusses the effect of back support on the transmission of vertical axis seat vibration to vertical axis at the head, and vertical axis seat vibration to fore-and-aft axis at the head. This is also presented in Griffin (1990, ‘Handbook of Human Vibration’) chapter 8. A reference and discussion of the two studies is required.
Line 491-2
The bracket opened on line 491 is not closed.
Line 557-561
Reference ‘Paddan and Griffin, 1988’ does not appear in the references list.
Line 627
Reference ‘Kitazaki and Griffin (1998)’ is dated 1997 in the references list.
Line 671
Change “… increasing in the …” to “… increasing the …”.
Line 685
Change “… is combined …” to “… is a combined …”.
Line 687
Reference ‘Kitazaki and Griffin (1998)’ is dated 1997 in the references list.
Line 708
Change “… assume lordotic …” to “… assume a lordotic …”.
Line 725
Change “… considerably reduction …” to “… considerable reduction …”.
Line 740
Change “… human sit …” to “… humans sit …”.
Line 762-3
The bracket opened on line 762 is not closed.
Line 772
Change “… 2002b …” to “… 2002 …”.
Line 796
Change “… increasing in …” to “… increase in …”.
Author Response

(The authors gave the same response as above.)

Reviewer 3 Report
This is an original and very thorough article investigating seat to head transmissibility (STHT). STHT is not a research area I have conducted work in so several of my comments a geared to explaining this work to readers, who like me, may not be experienced in the area.
In the introduction, I think you need to make it clear on why this work is being done, I have a comment in the introduction that addresses this.
In the methods, I think you need reiterate and explain why a rigid seat is used and why sitting without a backrest is tested, as these two seating scenarios do not occur in vehicles or heavy equipment. Finally, please explain why a white noise vibration signal is used rather than real WBV data from a vehicle. The random signal is artificial but must be needed for experimental control purposes.
The paper is long and needs to be shortened in my opinion, I am not sure if the journal has limits. The conclusion succinctly summarize what was identified in the study and the results and discussion should succinctly identify the same important findings. Currently, all finding are presented in the result and I feel the most important finding should be discussed. The discussion is long (~3000 words and I think it should be cut in half. I am used to research papers with around 7 tables and figures total and a limit of 4500 words. In other words, best when the research is concise.
Please see the specific comments in the PDF.

Author Response

(The authors gave the same response as above.)
